# Effect of Diets with the Addition of Edible Insects on the Development of Atherosclerotic Lesions in ApoE/LDLR^−/−^ Mice

**DOI:** 10.3390/ijms25137256

**Published:** 2024-07-01

**Authors:** Hayat Hassen, Petra Škvorová, Kshitiz Pokhrel, Martin Kulma, Ewa Piątkowska, Renata B. Kostogrys, Lenka Kouřimská, Tomasz Tarko, Magdalena Franczyk-Żarów

**Affiliations:** 1Department of Human Nutrition and Dietetics, Faculty of Food Technology, University of Agriculture in Krakow, A. Mickiewicz Av. 21, 31-120 Kraków, Poland; hayataragaw87@gmail.com (H.H.); ewa.piatkowska@urk.edu.pl (E.P.); renata.kostogrys@urk.edu.pl (R.B.K.); 2Department of Human Nutrition, Faculty of Chemical and Food Engineering, Bahir Dar Institute of Technology, Bahir Dar P.O. Box 26, Ethiopia; 3Department of Microbiology, Nutrition and Dietetics, Czech University of Life Sciences Prague, Kamýcká 129, 165 00 Praha-Suchdol, Czech Republic; skvorova@af.czu.cz (P.Š.); pokhrelkshitiz99@gmail.com (K.P.); kourimska@af.czu.cz (L.K.); 4Department of Zoology and Fisheries, Czech University of Life Sciences Prague, Kamýcká 129, 165 00 Praha-Suchdol, Czech Republic; kulma@af.czu.cz; 5Department of Fermentation Technology and Microbiology, Faculty of Food Technology, University of Agriculture in Krakow, A. Mickiewicz Av. 21, 31-120 Kraków, Poland

**Keywords:** *Gryllus assimilis*, *Tenebrio molitor*, antioxidant status, atherosclerosis, mouse model, lipid profile, fatty acids profile, liver enzymes activity

## Abstract

Foods enriched with insects can potentially prevent several health disorders, including cardiovascular diseases, by reducing inflammation and improving antioxidant status. In this study, *Tenebrio molitor* and *Gryllus assimilis* were selected to determine the effect on the development of atherosclerosis in ApoE/LDLR^−/−^ mice. Animals were fed AIN-93G-based diets (control) with 10% *Tenebrio molitor* (TM) and 10% *Gryllus assimilis* (GA) for 8 weeks. The nutritional value as well as antioxidant activity of selected insects were determined. The lipid profile, liver enzyme activity, and the fatty acid composition of liver and adipose tissue of model mice were evaluated. Quantitative analysis of atherosclerotic lesions in the entire aorta was performed using the *en face* method, and for aortic roots, the cross-section method was used. The antioxidant status of the GA cricket was significantly higher compared to the TM larvae. The results showed that the area of atherosclerosis (*en face* method) was not significantly different between groups. Dietary GA reduced plaque formation in the aortic root; additionally, significant differences were observed in sections at 200 and 300 µm compared to other groups. Furthermore, liver enzyme ALT activity was lower in insect-fed groups compared to the control group. The finding suggests that a diet containing edible insect GA potentially prevents atherosclerotic plaque development in the aortic root, due to its high antioxidant activity.

## 1. Introduction

The prevalence of cardiovascular diseases (CVDs) is increasing worldwide, and it is becoming a widespread public health issue [1]. One of the leading causes of cardiovascular diseases is atherosclerosis, an inflammation of the arteries associated with lipid and other metabolic alterations. 

Atherosclerosis is a progressive disease characterized by a gradual buildup of fatty deposits (plaque) within artery walls. This plaque narrows the arteries, hindering blood flow and potentially leading to serious complications like heart attack and stroke [2]. The development of atherosclerosis is linked to risk factors like high levels of low-density lipoprotein cholesterol (LDL) [3]. Furthermore, deficiencies in Apolipoprotein E (ApoE) and the LDL receptor (LDLR) are known to contribute to the disease. ApoE and LDLR play crucial roles in clearing LDL cholesterol from the bloodstream [4]. ApoE exerts anti-atherogenic effects through beneficial changes in plasma lipids and direct action on the artery. In ApoE/LDLR^−/−^ mice (animal model of atherosclerosis), a high level of plasma cholesterol is observed, which contributes to promoting the development of atherosclerotic plaque [4,5,6]. It has been indicated that instead of drug therapy, changing one’s diet and adopting a healthy lifestyle could prevent cardiovascular diseases, since diet significantly influences cardiovascular outcomes [7,8]. 

Evidence shows that consuming insects can reduce cytokines and modulate particular transcription factors to restore antioxidant and inflammatory status [9]. Edible insects provide various nutrients essential for human health, such as vitamins, minerals, fiber, protein, and lipids. These properties could enable them to improve gastrointestinal health, boost immunological function, decrease the risk of bacterial infection, and decrease chronic inflammation [10,11]. Studies on insects revealed decreased pro-inflammatory cytokine levels such as TNF-α (Tumor Necrosis Factor-alpha) in animals and humans. TNF-α is a critical pro-inflammatory cytokine linked to different diseases’ pathological processes [9,12,13]. 

Insects are a good source of crude fiber, most predominately in the form of chitin. Chitin and its degraded products have been shown to exert antimicrobial, antioxidant, anti-inflammatory, anticancer, and immunostimulatory activity [14,15]. 

A study in rats found that carbohydrate glycosaminoglycan derived from crickets showed a significant anti-inflammatory effect against chronic arthritis by lowering C-reactive protein (CRP) and rheumatoid factor, and thus suppressing a variety of inflammatory biomarkers [16,17]. Furthermore, previous research has shown that phytochemicals in mealworms have anti-obesity properties in mice. Obesity is one of the leading causes of inflammation in the body [18]. Additionally, creating biostimulants known as protein hydrolysates, edible insects, and foods enriched with insects may potentially prevent several health problems, including diabetes, hypertension, and cardiac issues [19]. 

Around the world, edible insects have been used as a food source, and in recent years, demand has grown in Europe. Within the European Union (EU), insects and insect-derived products are classified as novel foods and the manufacturing of insect products is governed by the Novel Foods Regulation (EU) 2015/2283, 25 November 2015 [20,21]. Placing freeze-dried and powder forms of *Tenebrio molitor* (TM, yellow mealworms) larvae on the market has been authorized by the Regulations (EU) 2021/882 and 2022/169. *Tenebrio molitor* and *Gryllus assimilis* (GA, Jamaican field crickets) are on the worldwide list of recorded edible insects [22].

Yellow mealworm larvae and Jamaican field crickets are gaining popularity as a potential source of nutrients and functional compounds. Research emphasizes their richness in essential amino acids, protein, and a diverse range of beneficial fatty acids [23,24,25]. The nutritional value of insects depends on many factors including rearing temperature, feed composition, strain, developmental stage, or sex [26]. Regarding the model species selected for this study, TM, belonging among the most researched edible insects, may vary in lipids 6.1–58.2% DM, 2.8–8.1% fiber DM, and proteins 38.9–76.2% DM. TM larvae reared on diets with a higher protein content tend to exhibit increased crude protein levels [27]. GA contains 62.7–65.5% of proteins, 20.8–33.0% of lipids, and 1.4–8.4% fiber in DM. It is also known that palmitic, oleic, and linoleic acids are the major fatty acid in both species [28,29,30,31]. TM larvae and GA cricket contain significant amounts of both monounsaturated fatty acids (MUFAs), and polyunsaturated fatty acids (PUFAs). Wu et al. [32] reported the percentage of MUFAs (47.35 ± 1.62%) and PUFAs (31.66 ± 0.99%) in freeze-dried mealworm larvae. Oleic acid, a major MUFA, is a key component (43.77 ± 1.62%) of TM larvae oil and exhibits well-documented anti-inflammatory properties [32]. Oleic acid exerts anti-inflammatory effects by reducing pro-inflammatory cytokines and promoting the production of anti-inflammatory mediators [33]. Furthermore, freeze-dried GA is a good source of MUFAs, containing 31.10 g/100 g of fat. Blanched GA contains PUFA at 35.12 g/100 g of fat [25]. Consuming both MUFAs and PUFAs has been linked to a lower risk of cardiovascular disease [34].

Studies have shown that insect hydrolysates and peptide fractions possess significant antioxidant activity, potentially reducing free radical levels and alleviating oxidative stress in the body. Previous work by Zielińska et al. [35] reported a particularly high antiradical activity against the free radical ABTS^•+^ in a hydrolysate derived from mealworms [35,36]. Additionally, TM larvae are a valuable source of bioactive compounds like phenolics, tocopherols, and chitosan. Protein and protein hydrolysates from mealworms have also been linked to anti-diabetic and anti-obesity effects [37,38]. A research study on GA identified a specific peptide within its hydrolysate that could potentially inhibit HMG-CoA reductase, an enzyme involved in cholesterol production, implying potential hypocholesterolemic properties. Crickets possess a high antioxidant potential, regardless of the analysis method used. This antioxidant activity might be attributed to the high tocopherol content found in cricket [39]. In addition to their antioxidant effects, edible insects exhibit anti-inflammatory properties. A review by D’Antonino et al. [9] highlights the ability of insect-derived compounds to reduce inflammatory cytokine production and modulate specific transcription factors, key contributors to inflammatory processes. 

The nutritional composition of TM and GA can also be influenced by processing techniques [25,40,41,42,43,44]. Freeze-dried TM larvae, in particular, have been shown to possess higher levels of phenolic compounds and antioxidant activities compared to other methods like oven drying and microwaving [45]. Research by Khatun et al. [25] demonstrated that freeze-dried GA crickets contain higher protein (66.63 g/100 g DM) and lower fat (21.19 g/100 g DM) compared to oven-dried and blanched samples. Other studies, such as those by Adamkova et al. [27] and Jozefiak et al. [46], found lower protein (~56 g/100 g DM) and higher fat (23–32 g/100 g DM), attributed to differences in the feed and developmental stages. 

While recent nutritional and environmental aspects of edible insects have been studied [47,48], data evaluating edible insects’ effects on the onset of atherosclerosis are limited. Therefore, the aim of the experiment was to describe the effect of *Tenebrio molitor* larvae (TM, yellow mealworms) and *Gryllus assimilis* (GA, Jamaican field crickets) on the development of atherosclerosis in apolipoprotein E/low-density lipoprotein receptor-deficient mice (ApoE/LDLR^−/−^) mice.

## 2. Results

### 2.1. Nutritive Value and Fatty Acid Composition of Insects 

The basic composition, including dry matter, ash, crude protein, fat, and chitin content was shown in Table 1. These components provide a thorough understanding of the nutritional value and structural attributes of the insects’ powder. Dry matter was similar in both insects’ powders, e.g., TM and GA. The amounts of fat and chitin were significantly lower in GA. Additionally, a significantly higher amount of crude protein was observed in GA compared to TM. 

The fatty acids profiles of insects’ powder are presented in Table 2. A higher level of SFAs, mainly 16:0 and 18:0, was observed in GA. Additionally, in GA, a higher amount of PUFAs, mainly octadecadienoic acid C18:2, was observed. However, *cis*-9-Octadecenoic acid (18:1) and finally MUFAs were lower compared to TM. 

### 2.2. Total Polyphenols Content and Antioxidant Activity of Insects’ Powder

The total polyphenols content of insects’ powder, including polyphenols in Gallic Acid Equivalents (GAE), is shown in Table 3. The antioxidant activity, expressed as ABTS^•+^ and DPPH^•^ radical scavenging activity percentages, and the Trolox equivalent antioxidant capacity (TEAC) are presented in Table 3. These metrics provide a comprehensive overview of the significant antioxidant potential of the insects. In GA, a higher antioxidant activity expressed as TEAC DPPH^•^ (μΜTrolox/1 g DM), as well as TEAC ABTS^•+^ (μΜTrolox/1 g DM), was observed. 

### 2.3. Composition of Experimental Diets

The standard AIN-93G (American Institute of Nutrition) rodent diet [49] was modified by replacing 10% of its content with freeze-dried insect powder. The detailed composition of these experimental diets is provided in Table 4.

### 2.4. Effect of Edible Insects on Body and Liver Weight and Biochemical Parameters in ApoE/LDLR^−/−^ Mice

There were no significant differences between experimental groups in body weight, fasting blood glucose, and plasma lipid profile after eight weeks of feeding (Table 5). The liver weight (g/100 g b.w.) was significantly higher in the GA group compared to the TM group. A significant decrease in liver enzyme activity ALT was also found in the group receiving GA and TM insects (Figure 1).

### 2.5. Effect of Edible Insects on Fatty Acids Composition of Adipose Tissue and Liver in ApoE/LDLR^−/−^ Mice

There were no changes in the proportion of saturated fatty acids (SFAs) between experimental groups in adipose tissue (Table 6). Mice fed a diet with TM had a significantly increased level of monounsaturated fatty acids (MUFAs) and decreased polyunsaturated fatty acids (PUFAs) compared to the control and GA groups. The palmitic acid (C16:0) level was significantly lower in the TM-fed group compared to the control. The linoleic acid (C18:2) level was significantly higher in the control and GA groups than in the TM-fed group.

The fatty acid profile of the liver showed a significantly higher share of SFAs in the GA group (Table 7). A significantly higher level of MUFAs was found in the TM group, whereas the level of PUFA was lower than in the other groups. There were no significant differences between the GA and the control group in relation to the proportions of MUFAs and PUFAs. Relatively higher levels of palmitoleic (C16:1), linoleic acid (C18:2), and α-linoleic (C18:3) acids were observed in the liver fatty acid profile of mice fed the control diet than the TM and GA groups. Furthermore, a significantly lower level of stearic acid (C18:0) was observed in the TM-fed group compared to the GA-fed group.

### 2.6. Effect of Edible Insects on the Development of Atherosclerosis in ApoE/LDLR^−/−^ Mice

After eight weeks of feeding ApoE/LDLR^−/−^ mice, no significant difference in the mean area of the atherosclerotic lesion (%) in the entire aorta measured by the *en face* method were observed (Figure 2a). The addition of GA to the diet led to a slight decrease in plaque levels in the mice. However, this decrease was not statistically significant.

The addition of edible insects (TM and GA) to the diet of ApoE/LDLR^−/−^ mice had no significant effect on the development of atherosclerosis counted as the total lesion area of the aortic root (Figure 3a). However, the analysis of individual cross-sections revealed that there was a significant (*p* < 0.05) difference in sections of the aortic root on the level of 200 and 300 µm far from the appearance of the first part of the aortic valve in the GA group compared to the control and TM groups (Figure 3c).

## 3. Discussion

The effect of edible insects (TM and GA) on the area of atherosclerotic lesions in ApoE/LDLR^−/−^ mice was studied. No significant changes in body weight or fasting blood glucose levels in mice fed diets containing edible insects (TM and GA) compared to the control group were observed. While previous research suggests that extracts from TM larvae can influence body weight, variations in insect type, origin, and diet could cause the observed differences [48,50].

Certain substances from insects, such as chitin from mealworms and glycosaminoglycans from crickets, have been shown to regulate lipid metabolism and blood lipid levels [16,47]. Also, it was shown that edible insects are a novel source of bioactive peptides. *Gryllus assimilis* was the only insect whose hydrolysis generated a peptide that was predicted to act as an inhibitor of the HMG-CoA reductase, thus suggesting a hypocholesterolemic property. In our study, it was observed that the amount of protein was significantly higher in GA compared to TM. Additionally, the cholesterol concentration as well as the LDL level tended to decrease in GA compared to TM. The present study did not observe changes in the plasma lipid profile between experimental groups; this represents a difference from previous findings on edible insects lowering blood cholesterol, triglycerides, and LDL levels [47,51]. 

Elevated serum ALT is associated with increased oxidative stress and systemic inflammation, critical components of atherosclerosis [52,53]. This study demonstrated that adding GA and TM into the diet of ApoE/LDLR^−/−^ mice could significantly reduce the activity of liver enzyme ALT but did not cause any difference in the serum activities of AST. The reduced activity of liver enzyme ALT might be caused by nutrients, antioxidants, and bioactive compounds found in edible insects [9,16]. The study by [54] reported that the defatted mealworm fermentation extract supplementation improved relative liver weight and reduced serum AST and ALT levels in alcohol-induced rats. Their findings imply that protein hydrolysates may benefit liver function [55,56]. Administering glycosaminoglycans (GAGs) extracted from field crickets was found to reduce oxidative enzyme levels, including ALT and AST. GAGs played a role in improving the metabolic activity of hepatocytes [57]. Insects are consumed as rich sources of protein, vitamins, minerals, fiber, and unsaturated fatty acids in some species [10].

The growing body of evidence highlights the potential benefits of PUFAs, especially n−3 PUFAs, for human health [58]. In our study, the amount of total PUFAs in adipose tissue and liver was higher in the GA group compared to the TM group. Our study also demonstrated that linoleic acid was significantly higher in the control and GA groups than in the TM-fed group. According to [59], the contents of unsaturated fatty acids in the field cricket were high, especially the oleic, linoleic, and linolenic acids. The fatty acid profile in adipose tissue indicated an increase in C18:1 and C18:2 (n−6) levels in the TM group compared to the control and GA groups. C18:1 and C18:2 (n−6) were the major fatty components of TM larvae [60]. 

The result from fatty acid analysis in the liver showed that the concentrations of MUFAs and PUFAs were significantly higher in the TM group, whereas the level of SFAs was significantly higher in the GA group. No significant difference in liver weight in mice fed experimental diets compared to the control was observed. Nevertheless, a significantly higher liver weight was observed in the GA group compared to the TM group. The elevated liver weight in the GA group may be due to the difference in fatty acid profiles of the insects. GA had a significantly higher SFA content (36.64%) compared to TM (33.84%). Studies suggest excessive SFA intake can promote liver fat accumulation, potentially contributing to a higher liver weight, whereas PUFAs prevent liver fat accumulation [61,62]. Furthermore, the lower MUFA content (35.27%) in GA crickets compared to TM larvae (41.19%) could also play a role, as MUFAs are generally considered beneficial for liver condition [63]. This difference might be caused by variations in the nutritional profiles of TM and GA, influencing the fatty acid assimilation and metabolism in the liver.

The 16:0/18:2 fatty acid ratio was calculated to assess the balance between palmitic acid and linoleic acid in liver and adipose tissue samples. In the liver, the TM group exhibited a notably higher 16:0/18:2 ratio than the control group, indicating an increased presence of palmitic acid relative to linoleic acid. Palmitate has been found to promote neointima formation by inducing inflammatory phenotypes in smooth muscle cells [64]. A study found that palmitate and stearate are proapoptotic, while non-esterified fatty acids, such as palmitoleate, oleate, and linoleate, exhibit anti-apoptotic properties by inhibiting NFκB activation in endothelial cells [65]. Interestingly, the GA group demonstrated a significantly lower 16:0/18:2 ratio in adipose tissue than the TM group. This result suggests a relatively higher proportion of linoleic acid than palmitic acid in the GA group, potentially indicating a more favorable fatty acid composition regarding inflammation and cardiovascular health [60,66]. 

It is mentioned that antioxidants combat oxidative stress, a key player in atherosclerosis development [67]. Despite no significant changes in total atherosclerotic lesion area, the incorporation of *Gryllus assimilis* into the diet has shown promising results in reducing atherosclerotic plaque areas in specific sections of the aortic root at 200 and 300 µm levels compared to the control and TM groups. This effect may be attributed to the strong antioxidant activity of these crickets. Recent studies have suggested that the consumption of edible insects can modulate oxidative stress due to their antioxidant potential [68]. While specific research on the antioxidant profile of GA is limited, broader studies on edible insects like *Acheta domesticus* (house cricket) larvae demonstrate the presence of various antioxidants, including phenolics [69,70]. In our study, we analyzed the antioxidant activity using various assays and we found that the ABTS^•^^+^ and DPPH^•^ expressed as Trolox equivalent antioxidant capacity (TEAC) assays showed the significantly higher values for *Gryllus assmilis*. 

Antioxidants are known to combat oxidative stress by neutralizing free radicals, reducing inflammatory cytokine production, and improving endothelial function [47]. The presence of phenolic compounds and other antioxidants in crickets has been well-documented [48,49,50]. Our findings support this with the notable amount observed in GA, which aligns with the observed health-related effects in our in vivo study. The radical scavenging activity observed in GA in our study suggests a potential mechanism for the observed health benefits. The high levels of antioxidants found in these crickets could explain the reduction in atherosclerotic plaque formation. The radical scavenging activity of GA, as observed in our study (75.75% for DPPH^•^), was higher compared to other cricket species such as *Acheta domesticus*, which showed a DPPH^•^ inhibition of 72%. This highlights the antioxidant properties of GA, further supporting their potential health benefits. The reduced oxidative stress might relate to the plaque reduction through several pathways. Antioxidants from GA may suppress inflammatory cytokine production and maintain healthy endothelial function; both crucial factors in preventing plaque buildup [71,72].

A research on nut bars fortified with cricket flour highlights the potential of edible insects as a source of antioxidants. This study found that cricket flour significantly increased the total phenolic content and tocopherol levels of the nut bars compared to standard bars [70]. Furthermore, another study reported reduced plasma TNF-α levels in healthy adults consuming cricket powder [12]. These findings suggest the potential anti-inflammatory effects of cricket’s antioxidants. 

Studies have also shown that insect glycosaminoglycans and unsaturated fatty acids can suppress inflammatory markers linked to cardiovascular disease development. Glycosaminoglycans derived from crickets can modulate the production of inflammatory mediators like IL-6 and PGE2, potentially contributing to an anti-inflammatory effect [14,15,16,17].

Phenolic compounds and other antioxidants present in GA crickets may contribute to this effect by reducing oxidative stress and inflammation, thereby improving cardiovascular and liver health and contributing to the development of therapeutic foods to prevent atherosclerosis. This promising finding, potentially linked to the anti-inflammatory properties of certain insect biomolecules, warrants further exploration. 

## 4. Materials and Methods

### 4.1. Insects

Both experimental species (*Tenebrio molitor*, *Linnaeus*, 1758, and *Gryllus assimilis*, *Fabricuis*, 1775) were obtained from the rearing facility of the Faculty of Agrobiology, Food and Natural Resources, Czech University of Life Sciences, Prague (FAFNR, CZU), and were maintained under conditions at 27 ± 1 °C and 40–50% relative humidity using a rack-system. The experimental crickets were kept in plastic rearing boxes (560 × 390 × 280 mm, SAMLA, IKEA, and Prague, Czech Republic) until their harvest at the age of 60 ± 1 days, when the majority of crickets were adults. The boxes were equipped with egg trays, two Petri dishes with feed and one Petri dish containing water gel (Oslavan, Náměšť nad Oslavou, Czech Republic). The mealworms were kept in plastic containers (280 × 140 × 390 mm, SAMLA, IKEA, Prague, Czech Republic) in feeding substrate when fresh sliced carrots were supplied as the only water source. TM were harvested using sieving when the first pupae occurred (the approximate age of harvested larvae was 90 days). 

Regarding the insects’ diet, both species were provided with dry feed ad libitum. GA larvae were provided with chicken feed (77.9% wheat, 17.6% soybean meal, 1.8% rapeseed oil, 2.7% premix of minerals, macronutrients, and micronutrients; particle size < 1 mm) produced in collaboration with the experimental farm of the Demonstrational and Experimental Centre, FAFNR, CZU. TM larvae were fed by wheat bran mixed with chicken feed (4:1). Before harvest, the experimental insects were starving for 24 h. After that, they were freeze-killed, lyophilized (Trigon Plus lyophilizer, Čestlice, Czech Republic), homogenized using a laboratory mill (A10; IKA Werke GmbH & Co. KG, Staufen, Germany), and stored at −80 °C.

#### 4.1.1. Nutritive Value and Fatty Acid Composition of Insects

Determination of basic composition of insect’s powder used as the feed component for experimental mice was carried out according to [73]. The dry matter (DM) was determined gravimetrically after drying the samples at 103 ± 2 °C (Memmert oven, Schwabach, Germany) to a constant weight. The samples were mineralized in a muffle furnace LAC (Verkon, Praha, Czech Republic) for ash content determination at 550 °C. The total fat content was determined using the Gerhardt Soxtherm SOX414 apparatus (C. Gerhardt GmbH and Co. KG, Königswinter, Germany). The crude protein content was evaluated with the Kjeltec 2400 analyzer (FOSS, Hilleroed, Denmark) using the nitrogen-to-protein conversion factor 6.25. Chitin was determined according to [36] after defatting and hydrolysis via 1M NaOH and 1M HCl. 

Fatty acid composition was analyzed using GC-MS (Shimadzu GC-MS, Model QP 5050A) by the method described previously [74].

#### 4.1.2. Determination of Total Polyphenols Content of Insects

The content of total phenolic compounds (TPC) of methanolic extracts was determined by the Folin–Ciocalteu (Sigma, St. Luis, MO, USA) colorimetric test developed by [69] with some modifications. The level of total polyphenolic compounds was determined spectrophotometrically at a wavelength of λ  =  760 nm using a RayLeigh UV-1800 spectrophotometer (Beijing Beifen-Ruili Analytical Instrument, Beijing, China). The results were expressed as g of Gallic Acid Equivalents (GAE) per 100 g of extract using a standard curve of gallic acid.

#### 4.1.3. Determination of Antioxidant Activity of Insects

##### ABTS^•+^ Radical Scavenging

Antioxidant activity was measured using the method of Re et al. with the ABTS^•+^ radical (2,2′-azino-bis(3-ethylbenzothiazoline-6-sulfonic acid) [75]. The absorbance of the colored solution was measured using a RayLeigh UV-1800 spectrophotometer (Beijing Beifen-Ruili Analytical Instrument, Beijing, China) at a wavelength of 734 nm in the presence of 70% methanol used to prepare the extracts. The result was expressed in μM Trolox/g DM.

##### DPPH^•^ Radical Scavenging

Antioxidant activity was determined by the method of Brand-Williams et al. using the free radical DPPH^•^ [76]. To a 1.5 mL sample suitably diluted with methanol, 3 mL of the prepared DPPH^•^ solution (diluted to an absorbance between 0.900 and 1.000) was added and the contents of the tube were mixed. The samples were incubated for 10 min and protected from light in room temperature. After this time, the absorbance was measured using a RayLeigh UV-1800 spectrophotometer at 515 nm against 99% pure undiluted methanol. The result was expressed in μM Trolox/g DM. The solution absorbance (A_sample_) was measured compared to the blank, in which the extract was replaced with methanol (A_blank_).

The percentage of DPPH^•^ inhibition was determined with the following equation:% inhibition = ((A_blank_ − A_sample_)/A_blank_) × 100

##### Ferric Reducing Antioxidant Power (FRAP)

Antioxidant activity was determined by the FRAP method according to Benzie and Strain [77]. The absorbance at 593 nm against 70% methanol was measured using a RayLeigh UV-1800 spectrophotometer (Beijing Beifen-Ruili Analytical Instrument Beijing, China). The results obtained were expressed in μM Trolox/g DM. The ABTS^•+^ radical scavenging activity was computed with the formula described in the section “DPPH^•^ radical scavenging activity assay”.

### 4.2. Animals and Housing

Animals (ApoE/LDLR^−/−^ mice) were bred in the Department of Human Nutrition and Dietetics, University of Agriculture, in Krakow. ApoE/LDLR^−/−^ mice are considered a suitable model to study the anti-atherosclerotic effect of treatments without an atherogenic diet since the animals develop hypercholesterolemia and atherosclerosis easily even when fed a standard chow diet [78,79]. This allows for observing the impact of dietary changes on a more established atherosclerotic process within a specific timeframe. ApoE plays a broader role in lipoprotein metabolism beyond just LDL uptake like LDLR [80]. According to [81], male animals appear to have more inflamed but smaller plaques than female animals.

Animals were housed in colony cages in constant environmental conditions (22–25 °C, 65–75% humidity) with a 12-h light/dark cycle and free access to diet and water. The body weight of mice was recorded weekly. All animal procedures were conducted according to the Guidelines for Animal Care and Treatment of the European Union and approved by the 1st Local Animal Ethics Commission in Krakow (Approval No. 562/2021, 28 October 2021).

### 4.3. Diets and Feeding

Mice were first fed a commercial cholesterol-free pelleted diet for two months (Sniff M-Z Spezialdiäten GmbH; Soest, Germany). Eight- to ten-week-old female mice (no signs of atherosclerosis) were used in the study. Animals were randomly assigned to three experimental groups (n = 10 in each group). The control group was fed the AIN-93G diet (American Institute of Nutrition) for the following eight weeks based on [49]. The second group was provided the AIN-93G diet supplemented with 10% of TM, and the third experimental group was fed the AIN-93G diet with 10% of GA for eight weeks.

### 4.4. Blood Sampling, Glucose, and Biochemical Analyses of Plasma

At the end of the nutritional experiment, mice were fasted for four hours and the blood glucose level from the tail vein was measured using Accu-Chek Active strips (Roche Diagnostics GmbH, Mannheim, Germany). Then, mice were injected intraperitoneally with heparin (1000 IU, Sanofi-Synthelabo; Paris, France) and after 10 min anaesthetised with ketamine/xylazine (20 μL, 100 mg/mL Biochemie; Vienna, Austria) and finally sacrificed by cervical translocation.

Blood was collected from the heart into tubes centrifuged at 14,000× *g* for 4 min to isolate plasma. The samples were kept at −80 °C for further analysis. ABX Pentra 400 biochemical analyzer (Horiba Medical, Japan) was used to perform the spectrophotometric method for determining the lipid profile, including total cholesterol level (TC), HDL and LDL cholesterol fractions and triacylglycerols (TAG), as well as the liver enzyme activities of alanine aminotransferase (ALT) and aspartate aminotransferase (AST) in the blood plasma.

### 4.5. Fatty Acids Profile of Adipose Tissue and Liver

At the end of the experiment, abdominal adipose tissues and liver were dissected from the mice, weighted and snap-frozen at −80 °C. Fat from the liver was extracted using a Leco TFE 2000 fat analyzer (Leco, St. Joseph, MO, USA) with liquid carbon dioxide as a solvent. The fatty acid profile was analyzed using GC-MS (Shimadzu GC-MS, Model QP 5050A) by the method described previously [74].

### 4.6. Quantification of Atherosclerosis in Aorta (En Face Method)

In anesthetized mice, the thorax was longitudinally opened, the right atrium was incised, and the heart was perfused in situ with phosphate-buffered saline (PBS, pH 7.4) through the apex of the left ventricle. The entire aortas (from the aortic arch to the iliac bifurcation) were dissected and kept in 4% buffered formalin for further analysis. The aortas were longitudinally opened and pinned to silicone plates to expose the surface for quantitative analysis of atherosclerotic lesions. Afterwards, aortas were stained with Sudan IV (Sigma-Aldrich, St. Louis, MO, USA) to visualize the atherosclerotic lesions as described previously [82]. Finally, the aortas were imaged, and the diameters of the lesions were quantified using LSM Image Browser 4 software (Zeiss, Jena, Germany).

### 4.7. Quantification of Atherosclerosis in Aortic Roots (Cross-Section Method)

For quantitative analysis of atherosclerotic lesions in the aortic root, the heart was cut and embedded in the OCT (optimal cutting temperature) compound (CellPath, Oxford, UK), then frozen and cut into serial cryosections of 10 μm thickness using a Cryostat CM1950 (Leica Biosystems, Deer Park, IL, USA). Eight sections were consecutively taken at intervals of 100 μm, beginning at 100 μm from the appearance of the aortic valves, following the protocol described previously [83]. Each section was mounted on slides and stained with oil-red-O (ORO) (Sigma-Aldrich, St. Louis, MO, USA) and examined under an Olympus BX50 (Olympus, Tokyo, Japan) microscope. A quantitative analysis of atherosclerotic lesions was performed using the LSM Image Browser 4 software (Zeiss, Jena, Germany). For each animal, a mean lesion area was calculated from nine sections and each section individually, reflecting the cross-section area covered by atherosclerosis.

### 4.8. Statistical Analysis

Obtained data are presented as the mean ± standard deviation (SD). The significance levels for basic composition, total polyphenol content, and antioxidant status of insects’ powder were determined using Student’s *t*-test. The normality of the results of in vivo study was determined using the Shapiro–Wilk test. The significance levels were determined using either the non-parametric Kruskal–Wallis test or a parametric test (one-way analysis of variance ANOVA with Tukey’s test) with the level of significance set at *p* < 0.05. All statistical analyses were conducted using STATISTICA 14.0.0.15 (TIBCO Software Inc., Palo Alto, CA, USA).

## 5. Conclusions

The addition of *Gryllus assimilis* to the diet appears to significantly decrease the atherosclerotic plaque area in sections of the aortic root, potentially due to its high antioxidant activity. Interestingly, our study found that adding GA to the diet significantly decreased the atherosclerotic plaque area in specific sections of the aortic root (200 and 300 µm) compared to the control group. To our knowledge, this is the first study to determine the effects of *Tenebrio molitor* and *Gryllus assimilis* on atherosclerotic lesion development in ApoE/LDLR^−/−^ mice.

The study does not include the composition of the atherosclerotic plaque, its stability, nor biomarkers of inflammation. It determined the effect of TM and GA only by quantifying the atherosclerotic plaque area. Further studies are needed to understand the underlying mechanisms, plaque composition, and stability. Additionally, clinical studies would be necessary in the future to prove the effect of insects on humans.

## Figures and Tables

**Figure 1 ijms-25-07256-f001:**
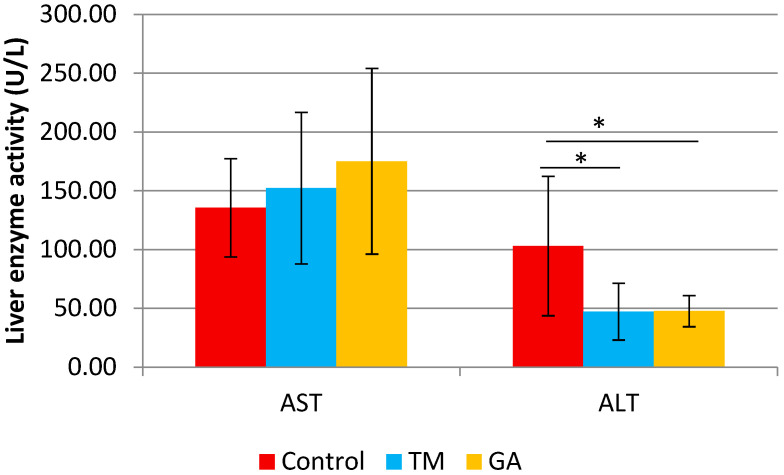
The liver enzymes alanine transaminase (ALT) and aspartate transaminase (AST) in ApoE/LDLR^−/−^ mice fed control, TM, and GA diets (n = 10). The mean difference is significant at *p* < 0.05 and indicated by *.

**Figure 2 ijms-25-07256-f002:**
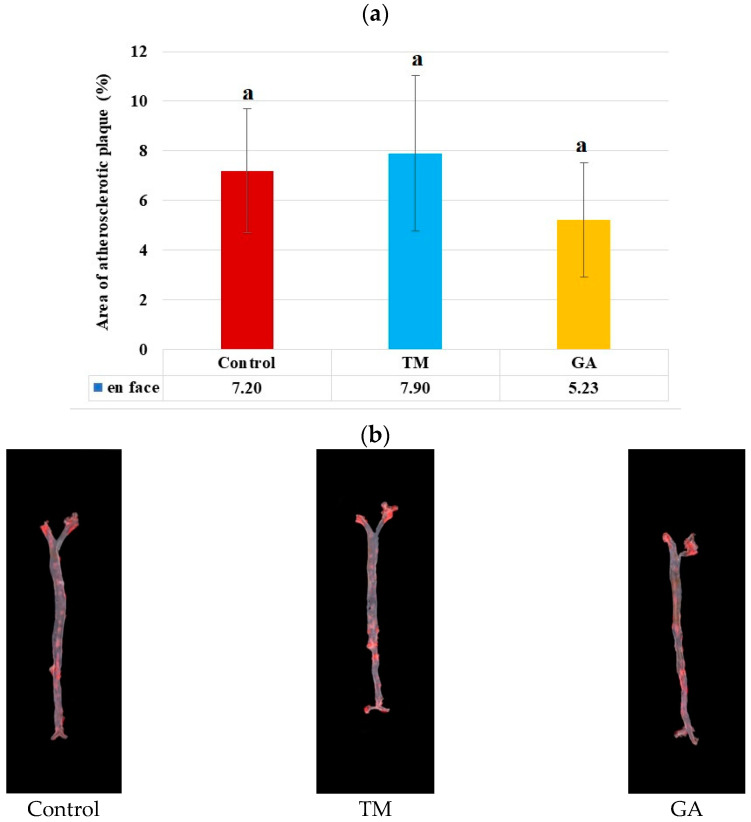
(**a**) Area of atherosclerotic lesions (%) in the entire aorta of ApoE/LDLR^−/−^ mice measured by the *en face* method (n = 10 for each group). The mean difference is significant at *p* < 0.05 and indicated by different letters. (**b**) Representative images of the aorta surface stained with Sudan IV. The red stained area indicates atherosclerosis plaques.

**Figure 3 ijms-25-07256-f003:**
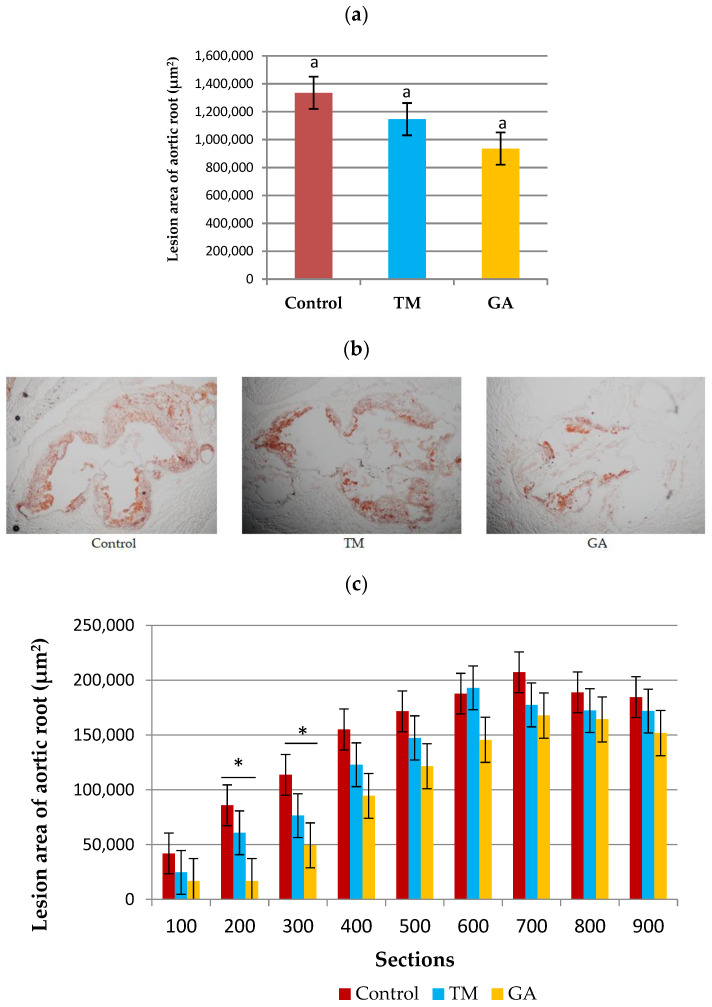
(**a**) The development of atherosclerosis in ApoE/LDLR^−/−^ mice (n = 10) measured by the cross-section method. (**b**) Representative images of cross-sections of aortic roots stained by ORO. (**c**) Sizes of the atherosclerotic lesion areas of nine individual cross-sections of the aortic root (n = 10 for each group). The mean difference is significant at *p* < 0.05 and indicated by different letters and *. Magnification of ×100.

**Table 1 ijms-25-07256-t001:** Basic composition of insects’ powder.

Insect	DM (g/100 g Insects’ Powder)	Ash (g/100 g DM)	Crude Protein (g/100 g DM)	Fat (g/100 g DM)	Chitin (g/100 g DM)
TM	96.19 ± 0.11	3.39 ^a^ ± 0.06	57.06 ^a^ ± 0.14	28.67 ^a^ ± 0.02	9.36 ^a^ ± 0.07
GA	96.10 ± 0.14	3.90 ^b^ ± 0.08	67.04 ^b^ ± 0.35	22.33 ^b^ ± 0.28	7.03 ^b^ ± 0.02

DM = dry matter, TM = *Tenebrio molitor* freeze-dried powder; GA = *Gryllus assimilis* freeze-dried powder. Data are presented as the mean ± SD. The mean difference is significant at *p* < 0.05 and indicated by different letters.

**Table 2 ijms-25-07256-t002:** Fatty acids composition of insects’ powder (in relative % of total fatty acids).

Systematic Name (IUPAC)	Number of Carbon Atoms: Number π Bonds	Insect	
		TM	GA
Dodecanoic acid	C12:0	0.46 ^a^ ± 0.02	0.19 ^b^ ± 0.08
Tetradecanoic acid	C14:0	5.58 ^a^ ± 0.27	1.81 ^b^ ± 0.89
Pentadecanoic acid	C15:0	0.21 ± 0.02	0.23 ± 0.05
Hexadecanoic acid	C16:0	18.84 ^a^ ± 0.30	22.16 ^b^ ± 1.53
*cis*-9-Hexadecenoic acid	C16:1	5.06 ± 0.22	4.65 ± 0.34
Heptadecanoic acid	C17:0	0.3 ± 0.05	0.27 ± 0.05
*cis*,*cis*-9,12-Hexadecadienoic acid	C16:2	0.89 ^a^ ± 0.02	0.41 ^b^ ± 0.25
Octadecanoic acid	C18:0	8.45 ^a^ ± 0.14	11.98 ^b^ ± 0.81
*cis*-9-Octadecenoic acid	C18:1	36.13 ^a^ ± 0.71	30.61 ^b^ ± 1.26
Octadecadienoic acid	C18:2	20.44 ^a^ ± 0.57	23.43 ^b^ ± 0.76
all *cis*-9,12,15-Octadecatrienoic acid	C18:3 (n − 3)	2.58 ^a^ ± 0.05	3.27 ^b^ ± 0.19
Other fatty acids		1.07 ± 0.17	0.98 ± 0.53
MUFA		41.19 ^a^ ± 0.92	35.27 ^b^ ± 1.51
PUFA		23.91 ^a^ ± 0.52	27.11 ^b^ ± 0.40
SFA		33.84 ^a^ ± 0.49	36.64 ^b^ ± 0.87

TM = *Tenebrio molitor* freeze-dried powder; GA = *Gryllus assimilis* freeze-dried powder; MUFA = monounsaturated fatty acid, PUFA = polyunsaturated fatty acid, SFA = saturated fatty acid. The mean difference is significant at *p* < 0.05 and indicated by different letters.

**Table 3 ijms-25-07256-t003:** Total polyphenols content and antioxidant status of insects’ powder.

Parameters	Insect	
	TM	GA
Polyphenols (mg GAE/100 g DM)	567.92 ± 34.05	568.85 ± 31.15
TEAC ABTS ^•+^ (μΜTrolox/1 g DM)	47.3 ^a^ ± 0.42	89.94 ^b^ ± 0.7
RSA ABTS ^•+^ (%)	77.3 ± 0.42	71.2 ± 0.28
TEAC DPPH^•^ (μΜTrolox/1 g DM)	81.52 ^a^ ± 0.92	178.63 ^b^ ± 1.46
RSA DPPH^•^ (%)	76.15 ± 0.49	75.75 ± 0.35
TEAC TPTZ (FRAP) (μmolTrolox/1 g DM)	65.27 ± 10.28	54.36 ± 7.18

GAE = Gallic Acid Equivalent, RSA = radical scavenging activity; TEAC = Trolox equivalent antioxidant capacity. The results are expressed as the mean ± SD. The mean difference is significant at *p* < 0.05 and indicated by different letters.

**Table 4 ijms-25-07256-t004:** Composition of experimental diets (g/kg).

Ingredient (g/kg)	AIN-93G (Control)	AIN-93G + 10% TM (TM)	AIN-93G + 10% GA (GA)
Casein	200	146	144
Protein from insect	-	54	56
Cornstarch	533	533	533
Sucrose	100	100	100
Cellulose	50	44	43
Chitin from insect	-	6	7
Soybean oil	70	34	53
Fat from insect	-	36	17
Vitamin mixture	10	10	10
Mineral mixture	35	35	35
Choline bitartrate	2.5	2.5	2.5
t-butylhydroquinone	0.014	0.014	0.014
TM	-	100	-
GA	-	-	100

TM = *Tenebrio molitor* freeze-dried powder; GA = *Gryllus assimilis* freeze-dried powder.

**Table 5 ijms-25-07256-t005:** Body weight (b.w.), liver weight, blood glucose, and plasma lipid profile in ApoE/LDLR^−/−^ mice fed control and diets with edible insects (TM and GA) (n = 10).

Parameters	Experimental Groups		
	Control	TM	GA
Body weight (g)	20.96 ± 1.37	20.39 ± 1.24	19.87 ± 1.99
Liver weight (g/100 g)	4.18 ^ab^ ± 0.32	3.86 ^a^ ± 0.44	4.39 ^b^ ± 0.48
Blood glucose (mg/dL)	132.30 ± 14.31	118.50 ± 22.01	132.82 ± 22.54
TC (mmol/L)	20.58 ± 2.20	22.75 ± 2.51	21.92 ± 2.74
LDL (mmol/L)	8.86 ± 1.01	9.14 ± 1.42	8.01 ± 2.21
HDL (mmol/L)	1.00 ± 0.13	0.85 ± 0.45	0.85 ± 0.18
TAG (mmol/L)	2.34 ± 0.58	1.85 ± 0.39	2.13 ± 0.47

TC = total cholesterol; LDL = low-density lipoprotein; HDL = high-density lipoprotein; TAG = triacylglycerols. Data are presented as the mean ± SD. The mean difference is significant at *p* < 0.05 and indicated by different letters.

**Table 6 ijms-25-07256-t006:** Fatty acid profile (in relative % of total fatty acids) of adipose tissue in ApoE/LDLR^−/−^ mice fed modified diets with edible insects.

Systematic Name (IUPAC)	Number of Carbon Atoms: Number π Bonds	Experimental Groups		
		Control	TM	GA
Dodecanoic acid	C12:0	0.15 ^a^ ± 0.02	0.20 ^b^ ± 0.01	0.14 ^a^ ± 0.03
Tetradecanoic acid	C14:0	2.45 ^b^ ± 0.21	2.85 ^c^ ± 0.20	2.16 ^a^ ± 0.26
Pentadecanoic acid	C15:0	0.15 ^a^ ± 0.03	0.22 ^b^ ± 0.05	0.13 ^a^ ± 0.02
14-Methylpentadecanoic acid	C16:0 iso	0.15 ^a^ ± 0.03	0.15 ^a^ ± 0.02	0.13 ^a^ ± 0.02
Hexadecanoic acid	C16:0	21.48 ^a^ ± 1.59	19.80 ^b^ ± 0.46	20.58 ^a,b^ ± 0.77
*cis*-9-Hexadecenoic acid	C16:1	3.30 ^a^ ± 0.68	4.43 ^b^ ± 0.61	3.21 ^a^ ± 0.56
Heptadecanoic acid	C17:0	0.37 ^a^ ± 0.08	0.51 ^b^ ± 0.07	0.44 ^a,b^ ± 0.08
*cis*-10-Heptadecanoic acid	C17:1	0.42 ^a^ ± 0.07	0.62 ^b^ ± 0.05	0.55 ^a^ ± 0.06
Octadecanoic acid	C18:0	12.42 ^a^ ± 1.60	11.70 ^a^ ± 1.61	13.25 ^a^ ± 1.67
*cis*-9-Octadecenoic acid	C18:1	29.03 ^a^ ± 0.84	32.96 ^c^ ± 0.46	30.41 ^b^ ± 1.09
Nonadecanoic acid	C19:0	0.13 ^a^ ± 0.09	0.20 ^a^ ± 0.06	0.21 ^a^ ± 0.08
*cis*,*cis*-9,12-Octadecadienoic acid	C18:2 (n − 6)	23.96 ^a^ ± 1.09	21.07 ^b^ ± 1.20	23.26 ^a^ ± 1.33
all *cis*-6,9,12-Octadecatrienoic acid	C18:3 (n − 6)	0.36 ^a^ ± 0.12	0.29 ^a^ ± 0.06	0.36 ^a^ ± 0.11
all *cis*-9,12,15-Octadecatrienoic acid	C18:3 (n−3)	4.72 ^a^ ± 1.46	4.32 ^a^ ± 0.43	4.53 ^a^ ± 0.56
*cis*,*cis*-11,14-Eicosadienoic acid	C20:2 (n − 6)	0.32 ^b^ ± 0.32	0.15 ^a,b^ ± 0.05	0.12 ^a^ ± 0.03
all *cis*-5,8,11,14-Eicosatetraenoic acid	C20:4 (n − 6)	0.13 ^a^ ± 0.06	0.14 ^a^ ± 0.03	0.11 ^a^ ± 0.03
Other fatty acids		0.33 ^a^ ± 0.17	0.29 ^a^ ± 0.17	0.37 ^a^ ± 0.08
Total SFA		37.30 ^a^ ± 1.42	35.64 ^a^ ± 1.41	37.04 ^a^ ± 1.75
Total MUFA		32.74 ^a^ ± 1.08	38.01 ^b^ ± 0.51	34.17 ^a^ ± 1.55
Total PUFA		29.63 ^a^ ± 1.37	26.06 ^b^ ± 1.17	28.43 ^a^ ±1.56
18:2/18:3 (n−6/n−3)		5.66 ^a^ ± 2.13	4.93 ^a^ ± 0.65	5.20 ^a^ ± 0.65
18:1/18:0		2.37 ^a^ ± 0.30	2.86 ^b^ ± 0.39	2.33 ^a^ ± 0.35
16:1/16:0		0.15 ^a^ ± 0.03	0.22 ^b^ ± 0.03	0.16 ^a^ ± 0.03
16:0/14:0		8.84 ^a^ ± 0.96	6.97 ^b^ ± 0.51	9.69 ^a^ ± 1.45
18:0/16:0		0.58 ^a^ ± 0.11	0.59 ^a^ ± 0.09	0.64 ^a^ ± 0.09
16:0/18:2		0.90 ^a,b^ ± 0.05	0.94 ^b^ ± 0.05	0.89 ^a^ ± 0.05

The mean difference is significant at *p* < 0.05 and indicated by different letters.

**Table 7 ijms-25-07256-t007:** Fatty acids profile (in relative % of total fatty acids) in the liver in ApoE/LDLR^−/−^ mice fed modified diets with edible insects.

Systematic Name (IUPAC)	Number of Carbon Atoms: Number π Bonds	Experimental Groups		
		Control	TM	GA
Dodecanoic acid	C12:0	0.06 ^a^ ± 0.03	0.05 ^a^ ± 0.03	0.06 ^a^ ± 0.03
Tetradecanoic acid	C14:0	1.05 ^a^ ± 0.43	0.90 ^a^ ± 0.22	0.91 ^a^ ± 0.13
Pentadecanoic acid	C15:0	0.15 ^a^ ± 0.03	0.17 ^a^ ± 0.06	0.15 ^a^ ± 0.03
12-Methylpentadecanoic acid	C16:0 iso	0.05 ^a^ ± 0.02	0.06 ^a^ ± 0.02	0.05 ^a^ ± 0.01
Hexadecanoic acid	C16:0	21.81 ^a^ ± 0.86	23.17 ^a^ ± 2.55	22.73 ^a^ ± 1.05
*cis*-9-Hexadecenoic acid	C16:1	3.84 ^b^ ± 1.33	3.19 ^a,b^ ± 1.03	2.69 ^a^ ± 0.60
*cis*,*cis*-9,12-Hexadecadienoic acid	C16:2	0.32 ^a^ ± 0.11	0.36 ^a^ ± 0.10	0.41 ^a^ ± 0.13
Heptadecanoic acid	C17:0	0.36 ^a^ ± 0.05	0.65 ^a^ ± 0.73	0.48 ^a^ ± 0.08
Octadecanoic acid	C18:0	9.81 ^a,b^ ± 0.89	9.42 ^a^ ± 1.25	10.77 ^b^ ± 1.02
*cis*-9-Octadecenoic acid	C18:1	28.80 ^a^ ± 1.16	31.76 ^b^ ± 0.77	28.58 ^a^ ± 1.05
*cis*,*cis*-9,12-Octadecadienoic acid	C18:2(n − 6)	23.16 ^b^ ± 1.99	21.28 ^a^ ± 1.13	22.92 ^a,b^ ± 1.40
all *cis*-6,9,12-Octadecatrienoic acid	C18:3 (n − 6)	1.07 ^a^ ± 0.28	0.84 ^a^ ± 0.25	0.95 ^a^ ± 0.18
all *cis*-9,12,15-Octadecatrienoic acid	C18:3 (n−3)	3.22 ^b^ ± 0.47	2.55 ^a^ ± 0.79	3.09 ^a,b^ ± 0.45
*cis*,*cis*-11,14-Eicosadienoic acid	C20:2 (n − 6)	0.74 ^a^ ±0.26	0.60 ^a^ ± 0.17	0.66 ^a^ ± 0.16
all *cis*-5,8,11,14-Eicosatetraenoic acid	C20:4 (n − 6)	3.07 ^a^ ± 0.64	3.00 ^a^ ± 0.62	3.18 ^a^ ± 0.94
all *cis*-4,7,10,13,16-Docosapentaenoic acid (DPA)	C22:5 (n − 6)	0.21 ^b^ ± 0.10	0.15 ^a,b^ ± 0.08	0.10 ^a^ ± 0.04
all *cis*-4,7,10,13,16,19-Docosahexaenoic acid (DHA)	C22:6 (n − 3)	1.48 ^a^ ± 0.33	1.28 ^a^ ± 0.35	1.49 ^a^ ± 0.54
Other fatty acids		0.49 ^a,b^ ± 0.14	0.37 ^a^ ± 0.15	0.55 ^b^ ± 0.17
Total SFA		33.29 ^a^ ± 1.62	34.40 ^a,b^ ± 1.77	35.14 ^b^ ± 1.28
Total MUFA		32.64 ^a^ ± 1.89	34.95 ^b^ ± 1.37	31.27 ^a^ ± 1.18
Total PUFA		33.58 ^a^ ± 2.27	30.28 ^b^ ± 1.70	33.04 ^a^ ± 1.92
18:2/18:3 (n-6/n-3)		7.35 ^a^ ± 1.41	11.44 ^a^ ± 11.91	7.55 ^a^ ± 1.14
18:1/18:0		2.96 ^a^ ± 0.33	3.44 ^b^ ± 0.56	2.68 ^a^ ± 0.31
16:1/16:0		0.18 ^b^ ± 0.06	0.14 ^a,b^ ± 0.05	0.12 ^a^ ± 0.02
16:0/14:0		23.98 ^a^ ± 10.11	28.68 ^a^ ± 15.16	25.28 ^a^ ± 2.43
18:0/16:0		0.45 ^a^ ± 0.04	0.41 ^a^ ± 0.09	0.48 ^a^ ± 0.05
16:0/18:2		0.95 ^a^ ± 0.09	1.09 ^b^ ± 0.14	0.99 ^a,b^ ± 0.07

The mean difference is significant at *p* < 0.05 and indicated by different letters.

## Data Availability

Data will be available upon request from the corresponding authors via email: magdalena.franczyk-zarow@urk.edu.pl.

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
