# Peer review of "Effect of Diets with the Addition of Edible Insects on the Development of Atherosclerotic Lesions in ApoE/LDLR^−/−^ Mice"

_ijms, 2024, doi:10.3390/ijms25137256_

Round 1
Reviewer 1 Report
Comments and Suggestions for Authors
The authors evaluate incorporating tenebrious and crickets into the diet to reduce atherosclerotic lesions.
I think it is necessary to focus the introduction on the described characterization of these insects in terms of their nutritional value and not speak in general terms of “insects” because it depends on the insect (Coleoptera or Orthoptera) in the stage of development of the insect used: larva (mealworm), molted worm, large worm, pupae or young or adult beetle.
The nutritional characterization of tenebrious has already been described in other studies:
1. https://link.springer.com/chapter/10.1007/978-3-030-32952-5_20#:~:text=The%20live%20mealworm%20is%20made,have%20a%20profitable%20protein%20profile
2. https://www.mdpi.com/2076-3921/11/9/1840
Discuss the differences in procedure and nutrient composition (among them, the concentration of polyphenols).
Specific questions:
+ Why does soybean oil content change in insect diets?
+ Explain why there is a significant increase in liver weight in the GA group
+ Please add the meaning of the letters a, b, B, which are included in graphs and tables. They are not easy to interpret. In graphs, I suggest only marking significant differences with a horizontal bar that joins the compared groups and the P value
+ Regarding the activity of liver enzymes, a control group should be included to identify the metabolic alterations of the knockout mouse and contrast it with the different diets. Contrast the results of the knockout mouse compared to other articles that evaluate liver enzymes because they are not very similar. For example: https://www.nature.com/articles/labinvest2014112
+ There appears to be a modest effect in an area of aortic root injury, although the images of the GOLD-stained tissues are not very evident or are not the best photographs.
Author Response
Manuscript ID: ijms-3060524
Title: Effect of Diets with the Addition of Edible Insects on the Development of Atherosclerotic Lesions in ApoE/LDLR-/- Mice
Please find our manuscript revised according to Reviewer’s comments
General remarks:
Thank you very much for the valuable feedback. We have addressed all the raised queries in our response below. Additionally, we have revised the manuscript based on the comments and submitted the final version to the system.
Reviewer #1:
Comment: In the introduction on the described characterization of these insects in terms of their nutritional value and not speak in general terms of “insects”.
Response: The detailed information about the nutritional value and health-related effects of specific insects has been added to the introduction’s section (Line 89-123).
Comment: Discuss the differences in procedure and nutrient composition (among them, the concentration of polyphenols).
Response: According to reviewer’s suggestion the impact of processing techniques on nutritional composition of studied insects has been included to the introduction’s section (Line 124-132).
Specific questions:
Comment: Why does soybean oil content change in insect diets?
Response: There were not changes of soybean oil content in insect diet, but the changes of soybean oil amount in the experimental diets of model mice (Table 4). No soybean oil was added to the feed for insects. We adjusted the amount of soybean oil in mice diet based on the content of fat in the specific insect.
Comment: Explain why there is a significant increase in liver weight in the GA group.
Response: The explanation has been added to the discussion’s section (Line 345-356).
Comment: Please add the meaning of the letters a, b, B, which are included in graphs and tables. They are not easy to interpret. In graphs, I suggest only marking significant differences with a horizontal bar that joins the compared groups and the P value.
Response: The meaning of the significant differences (p<0.05) shown as various letters and horizontal bars with asterisks are explain below the Figures and Tables.
Comment: Regarding the activity of liver enzymes, a control group should be included to identify the metabolic alterations of the knockout mouse and contrast it with the different diets. Contrast the results of the knockout mouse compared to other articles that evaluate liver enzymes because they are not very similar.
Response: The literature indicated by Reviewer (Kampschulte et al., 2014) reports elevated levels of liver enzyme ALT, when single knockout ApoE-deficient mice were fed a high-fat, high-cholesterol Western diet. This research uses the Western diet, which is different from the diets used in our study. Dietary components can significantly influence liver enzyme activity (Li et al., 2024, Linkon 2023). The main aim of our experiment was to compare the effect of a diet supplemented with edible insects on the development of atherosclerosis in double knockout mice (ApoE/LDR-/-), therefore the mouse model of atherosclerosis was used in this experiment. The role of diet was not investigated in control C57BL/6J mice.
Comment: There appears to be a modest effect in an area of aortic root injury, although the images of the GOLD-stained tissues are not very evident or are not the best photographs.
Response: We have replaced the figures with other images to improve clarity.
Reviewer 2 Report
Comments and Suggestions for Authors
The authors draw attention to insect-enriched foods as potential protectors of cardiovascular disease by presenting information on the anti-inflammatory effect of Tenebrio molitor (TM) and Gryllus assimilis (GA) in the development of atherosclerosis in ApoE/LDLR-/- mice. The use of insects, insect meals as protectors of various diseases will expand in the next 5 years. The manuscript is inovative, written well, and present this sort of studies for the first time.
Minor comments:
1. Abstract- part 3 and 4 to be rewritten; Graphical abstact is need;
2. In introduction part - to be add a brief explanation of atherosclerosis as a disease type, atherosclerosis development and in apolipoprotein E/low density lipoprotein receptor (ApoE/ LDLR-/-) deficiency in mice.
In introduction part -the finding is raised on CRP, pro-inflammatory cytokines, TNF ... are not mentioned in the results and discussion; I suggest rewriting the introduction by tracking the changes in antioxidant activity, the action of PUFA, MUFA ...... and then the changes according to literature data of liver markers after the use of insects.
3. Materials and Methods
Basic composition determination of insects and Fatty acids composition of insects it would be better if they were to unite;
Determination of total polyphenols content and antioxidant activity of insects- it would be good to separate them; to indicate the TPC first; and in AA all methods should be described briefly.
row 367-372 to be shorten
fed AIN-93G diet -to give a full explanation after the abbreviation
In matherials and methods section no mention the measuring body weightweight, Liver weight... and in witch days was made ... directly mentioned in results!? To be add.
4. Results
Table 1, table 2 are well presented
Table 3 TEAC ABTS have to be TEAC ABTS.+ ; for DPPH. methods use the radical scavenging, nor the molecule
table 4. Results were expressed as mean ± SD. The means bearing different letters were significantly 110 different (p < 0.05). to be add.
Figure 1 and table 6 are in italic - to unify the style
Figure 3a it will be good to submit to statistics and Figure 3a and 3c is possible to be combined. Figure 3b to be add agin -it is not effective
Figure 3c atherosclerotic lesion areas - have to present the resolution, x 100, x 400 ....
5. Discussion
This part is poor. To be re-write, by following the connection antioxidants- radical scavenging, oxidative stress-protecting liver markers, glucose levels and atherosclerosise un-deposition. In discussion part no evidence/ecplanation forSizes of the atherosclerotic lesion areas hysthopatological explanations.
The limitation and future prospects of the study is not mention.
References - most of the uced references are from the last 5 years (49%).
Comments on the Quality of English Language
-
Author Response
Manuscript ID: ijms-3060524
Title: Effect of Diets with the Addition of Edible Insects on the Development of Atherosclerotic Lesions in ApoE/LDLR-/- Mice
Please find our manuscript revised according to Reviewer’s comments
General remarks:
Thank you very much for the valuable feedback. We have addressed all the raised queries in our response below. Additionally, we have revised the manuscript based on the comments and submitted the final version to the system.
Reviewer #2:
Comment: Abstract- part 3 and 4 to be rewritten; Graphical abstract is need;
Response: We have rewritten and corrected parts of the abstract in accordance with the Reviewer’s suggestions. Graphical abstract was included.
Comment: In introduction part - to be add a brief explanation of atherosclerosis as a disease type, atherosclerosis development and in apolipoprotein E/low density lipoprotein receptor (ApoE/ LDLR-/-) deficiency in mice.
Response: According to Reviewer’s comment the introduction’s part has been completed with missing information (Line 48-58).
Comment: In introduction part -the finding is raised on CRP, pro-inflammatory cytokines, TNF ... are not mentioned in the results and discussion; I suggest rewriting the introduction by tracking the changes in antioxidant activity, the action of PUFA, MUFA ...... and then the changes according to literature data of liver markers after the use of insects.
Response: As required we have added additional information on this issue in the introduction’s part (Line 89-123).
- Materials and Methods
Comment: Basic composition determination of insects and Fatty acids composition of insects it would be better if they were to unite;
Nutritive value considering the fatty acid composition of insects have been combined into one sub-chapter.
Comment: Determination of total polyphenols content and antioxidant activity of insects- it would be good to separate them; to indicate the TPC first; and in AA all methods should be described briefly.
Response: According to Reviewer’s suggestion we have corrected these sub-chapters.
Comment: row 367-372 to be shortened
Response: As required we have shortened the paragraph.
Comment: fed AIN-93G diet -to give a full explanation after the abbreviation
Response: This information has been added. To maintain conciseness, we propose using the abbreviation (AIN-93G) followed by a brief in-text reference to a readily available publication that defines the diet (Reeves et al., 1993).
Comment: In matherials and methods section no mention the measuring body weight, Liver weight... and in which days was made ... directly mentioned in results!? To be add.
Response: As suggested the missing information has been included.
- Results
Comment: Table 3 TEAC ABTS have to be TEAC ABTS.+ ; for DPPH. methods use the radical scavenging, nor the molecule
Response: Thank you very much for the comment. We decided to present the results of the DPPH analysis in the same way as the ABTS in order to make it easier to compare the values obtained. Superscripts have been added.
Comment: table 4. Results were expressed as mean ± SD. The means bearing different letters were significantly 110 different (p < 0.05). to be add.
Response: Table 4 describes the composition of experimental diets for model mice. There are not the results from any measurements which could be subjected to any statistical evaluation. Therefore, the information about the data expression and indication of significant differences is missing there.
Comment: Figure 1 and table 6 are in italic - to unify the style
Response: As required the mistake has been corrected.
Comment: Figure 3a it will be good to submit to statistics and Figure 3a and 3c is possible to be combined. Figure 3b to be add agin -it is not effective
Response: Statistical analysis was presented at Figure 3a as letters – there were no significant differences. Figure 3c has been replaced with statistics presented as bars with asterisk.
Comment: Figure 3c atherosclerotic lesion areas - have to present the resolution, x 100, x 400
Response: As required the missing information at Figure 3b was added. Magnification of ×100
- Discussion
Comment: This part is poor. To be re-write, by following the connection antioxidants- radical scavenging, oxidative stress-protecting liver markers, glucose levels and atherosclerosise un-deposition. In discussion part no evidence/explanation for sizes of the atherosclerotic lesion areas hysthopatological explanations.
Response: According to Reviewer’s suggestions Discussion section has been re-written.
Comment: The limitation and future prospects of the study is not mention.
Response: This information has been added in the conclusion part (569-573).
Round 2
Reviewer 2 Report
Comments and Suggestions for Authors
-